# Developing and Piloting a Novel Ranking System to Assess Popular Dietary Patterns and Healthy Eating Principles

**DOI:** 10.3390/nu14163414

**Published:** 2022-08-19

**Authors:** Ella L. Bracci, Rachel Milte, Jennifer B. Keogh, Karen J. Murphy

**Affiliations:** 1Clinical and Health Sciences, Alliance for Research in Exercise, Nutrition and Activity, University of South Australia, GPO Box 2471, Adelaide, SA 5001, Australia; 2Caring Futures Institute, College of Nursing and Health Sciences, Flinders University, Adelaide, SA 5042, Australia

**Keywords:** weight loss diets, Australian Guide to Healthy Eating, nutrition profile, diet analyses, behaviour change

## Abstract

A multitude of weight loss diets exist. However, no one diet has been proven to be superior, despite their claims. Resultingly, this creates confusion amongst consumers and conflicting nutrition messages. The aim of the ranking system was to evaluate a range of dietary pattern’s nutrition profile and financial costs, as well as their potential long-term sustainability and associated adverse effects. Nutrition profile is typically the focal point of weight loss diets with less attention focused towards other factors that may affect their suitability. Five popular diets (Keto, Paleo, Intermittent Fasting, Optifast, and 8 Weeks to Wow) and two energy restricted healthy eating principles (Australian Guide to Healthy Eating and the Mediterranean Diet) were compared for diet quality, cost, adverse effects, and support for behaviour change. In general, healthy eating principles scored more favourably compared to popular weight loss diets in all categories. Lower carbohydrate diets tended to score lower for diet quality due to restricting multiple food groups, had more associated adverse effects and did not encourage behaviour change compared to the other weight loss diets. Optifast was the only weight loss diet to receive a negative score for cost. There should be considerations when undertaking a change to dietary patterns beyond nutrition profile. Diets indeed vary in terms of diet quality, and in addition can be costly, incur adverse effects, and disregard behaviour change which is important for sustainable weight loss and maintenance. This ranking system could create a reference point for future comparisons of diets.

## 1. Introduction

Long-term successful weight loss is often predicted by positive behaviour change to diet and lifestyle [1]. Without behaviour change, which may be overlooked in weight loss diets and short-term dieting, there is often weight gain post-diet and an increased risk for future obesity and weight cycling [2,3]. Fad diets are popular weight loss diets which peak in popularity (for example, the Dukan Diet, Atkins Diet, Zone Diet, and Skinny Teas), and are then replaced by the newest trends such as Keto, Paleo, and Intermittent Fasting [4,5]. Generally, fad diets do not promote positive behaviour change, can be costly, can result in negative and adverse symptoms, and are unsustainable for the long term [4,6,7]. Further, these diets often cut out food groups which can result in a less favourable nutrition profile and potential issues with micronutrient deficiency. The health claims that surround fad diets are concerning and promote unrealistic expectations [5]. Further, these diets typically lack scientific or evidence-base origin [4,5] compared to the Australian Guide to Healthy Eating (AGHE) and the Mediterranean Diet (MedDiet) which have a role in disease prevention and promote healthy ageing [2].

Although diets such as Keto and Intermittent Fasting can induce weight loss and have been previously utilized for treatment of medical disorders such as seizures, there is no evidence to the authors knowledge that one specific dietary pattern is superior to another for weight loss due to individual variability [8,9]. In fact, various studies have demonstrated that intermittent fasting, a popular tool for weight loss, yields similar weight loss and maintenance results compared to continuous fasting with an energy deficit [10,11,12].

Additionally, there is currently no index or tool to compare weight loss diets. Validated dietary assessment tools such as the Food Frequency Questionnaires (FFQ), Dietary Guideline Index (DGI), Diet Quality Index (DQI), and Healthy Eating Index (HEI) [13,14,15,16] provide insight of dietary quality, but could be built upon to provide further detail. For example, the Dietary Guideline Index (DGI) includes components such as water intake and physical activity which may not be accessible in all meal plan and weight loss resources [13]. Further, the associated scores from the DGI are not based on gender or life stage, but on food specifications [13]. This includes a score associated with the inclusion of low-fat milk as opposed to whole-fat milk, and never or rarely salting food compared to usually salting food [13]. Further, previous research that compares or assesses diets in relation to one another often only reports one finding, such as weight loss efficacy or cardiometabolic risk, as opposed to multiple outcomes.

Moreover, little is known about the cost of popular weight loss diets individually and compared to healthy eating guidelines such as the Australian Guide to Healthy Eating and the Mediterranean Diet. It is thought that healthy eating is costly and unobtainable [17] despite fad diets often requiring the purchase of expensive food alternatives such as low carbohydrate replacements or protein supplements. However, research suggests that consuming a healthy diet from the five core food groups can be more affordable than fad diets [18]. The current pandemic climate has highlighted the need for research regarding the cost of dietary patterns and the relationship with food insecurity in Australia. From 2019 to 2020, the proportion of Australian’s seeking food relief from organisations such as FoodBank doubled and many individuals were experiencing food insecurity for the first time [19,20]. Accompanying the lack of data on the cost of contemporary weight loss diets, few studies report the potential negative side effects or adverse effects that may occur with certain dietary patterns. Weight loss diets can result in symptoms such as dizziness, nausea, and lethargy, which are seldom discussed [21,22,23,24].

The aim of this research is therefore to develop and pilot a comprehensive ranking system to compare diets beyond nutritional profile. The ranking system will evaluate popular diets in terms of diet quality, financial costs, behaviour change, and adverse effects. Diets will be assessed through food group and micronutrient intake, cost in relation to income and food insecurity, encouragement of changing behaviour, and potential negative symptoms.

## 2. Materials and Methods

A combination of dietary software data, grey literature, and black literature were sourced to inform the ranking system. Popular diets were selected through methodology previously published by the authors [25]. In brief, Google Trend data and grey literature were used to ascertain the most popular diets in early 2019. Where diets were similar, for example, Optifast and Optislim, the more popular diet was selected based on relative search volume as per Google trend. Dietary input, analyses of nutrition profile, and cost methodology has also been previously published by [18]. Diets were scored as per the scoring templates previously published by the authors and can be found in the Appendix A.

Literature reviews were conducted through various databases to gather information regarding available dietary assessment tools, the cost of weight loss diets, requirements for behaviour change, and adverse effects related to weight loss diets. Various search terms and strategies, i.e., truncations, Boolean operators, and pearling, were used. Due to the nature of broader research questions and topics, and considerable differences in methodology, systematic reviews and meta-analyses were the preferred sources of information and eligibility criteria were less stringent. Two stages of screening were undertaken. Firstly, article titles and abstracts were screened to determine relevance for inclusion. If the article was deemed relevant to the topic, i.e., cost of weight loss, dietary assessment tools, or behaviour change requirements, the article was read in full text and added to Endnote referencing software. Articles were eligible when peer reviewed, content was relevant, full text was available, were written in English or had the option for full translation, and were relatively contemporary (<20 years old).

Grey literature including statistics, government documents from key agencies, policy statements, and practice guidelines were also used as part of the literature review. Pearling methods were also used to identify further resources of interest. Search tools included Google Scholar, the Australian Bureau of Statistics, Web of Science, trove, and government websites, i.e., Eat for Health, Queensland Government, the Cancer Council, and more. Similar search strategies were employed to gather information. Specific searches and search tools were used to locate relevant PDF documents, i.e., the American Dietary Guidelines and information sheets.

### 2.1. Nutritional Adequacy

Previous dietary assessment tools have reported macronutrient distribution, food group intake, and micronutrient intake to determine the diet quality of a population [13,26]. However, no one assessment tool was considered appropriate in the current research. To the knowledge of the authors, diet quality assessments have rarely been applied to weight loss or popular diets to assess their nutrition profile. Rankins et al., (2006) [15] evaluated various diets for nutritional adequacy in terms of macronutrient distribution (AMDR), essential fatty acids, micronutrients, and energy. This previous research suggested the plausibility of integrating and combining markers of diet quality and provided room for improvement to address supplementary information including gender differences.

Due to the AGHE as a comparator diet and the relevant healthy eating principle in an Australian context, diets were assessed and scored based on intake of the five food groups, AMDR, and micronutrient intake. Sub-categories were allocated 10 points if the requirements were met (gender, discretionary servings, AMDR, and saturated fat), whereas for other categories, the numerical value related to the number of servings per day, or meeting the recommended dietary intake (RDI) or estimated average requirement (EAR). For example, for vegetable intake 1 servings per day equated to 10 points, 2 servings were equal to 20 points, and so forth with a maximum score of 50 (100% of recommendations). With regards to micronutrients, if the average weekly intake met the relevant nutrient reference value (NRV), i.e., RDI or adequate intake (AI), it was appointed 20 points followed by 10 points for the EAR due to the lower value of the EAR. A full description and the scoring template are available within the Appendix A for reference.

### 2.2. Behaviour Change

An informal scoping review was conducted to identify key determinants of behaviour change and how weight loss and weight maintainers are successful compared to those that may be unsuccessful. While reviewing the literature, including the National Weight Control Registry (NWCR), similar beliefs and common practices were recognized as barriers or enablers to weight loss and weight maintenance [27,28,29]. Key foundations to successful weight loss included having healthy foods accessible, self-monitoring (weighing and food diaries), and decreasing access to high energy dense foods, [27,29]. Individuals who struggled with weight loss and maintenance (denoted as ‘Cluster 2′ in Ogden et al., 2012) had higher instances of weight cycling, or yoyo dieting, poorer health, and felt an increased effort in order to lose weight [29].

Similar to Rankins et al., (2006) [15], scoring the behaviour change category was problematic due to missing or impartial information. As a result, diets received a lower score contribution, i.e., 5 points per positive sub-category. A score of zero was allocated to diets that did not encourage positive behaviour change, i.e., did not provide a meal plan, had minimal support and resources, promoted yoyo dieting, and had food restrictions. The maximum possible positive score was 35 which indicated that the diet had positive alignment with guidelines and may have a role in long term weight management.

### 2.3. Cost

Previous research has calculated the cost of popular diets and weight loss diets [15,30,31,32,33]. However, diets have infrequently been allocated a score, or compared to other diets to evaluate cost-effectiveness. Rankins et al., (2006) [15] calculated the cost of weight loss menus per meal and per day for Atkins, Dietary Approaches to Stop Hypertension (DASH), Eat More Weight Loss, and South Beach diets. The total cost per day ranged from $7.91–$11.80 [15]. The DASH diet was considered the most economical as the meal plan had the lowest meal cost per energy basis (0.49 cost/Kcal) and overall cost per day ($7.91) [15]. Resultingly, the DASH diet was the only diet to receive an economic affordability score [15].

Though the literature has previously reported the cost of diets per day or per meal, in the current research, to assess the economic viability, diets were scored based in relation to food security and income. At any one time, up to 20% of the Australian population may experience food insecurity whereby access to sufficient, affordable, and nutritious food is restricted [20,34,35]. Spending greater than 25% of disposable income can increase the risk of food insecurity; however, for low-income earners, a healthy diet costs up to 30% of disposable income.

Diets therefore received a positive score of 10 points if the cost of weekly grocery shop was below the threshold of 30% of the average weekly income of an adult full-time worker in Australia ($1659 per week before tax) [36]. A negative score (−10 points) was allocated (e.g., Optifast) if the cost of groceries exceeded 30% of income.

### 2.4. Negative Symptoms

A literature review was conducted to identify the potential for short- and long-term health consequences of the diets, in addition to any noted adverse effects from each diet’s respective resource. Various databases were used including ProQuest, MDPI, Elsevier, BioPsychoSocial Medicine, and Springer. Common side effects ranged from headaches and nausea to GI disturbance such as diarrhea and constipation and were therefore included in the ranking system.

Assigning a score proved difficult due to insufficient evidence for certain diets such as the healthy eating principles (AGHE and MedDiet). Although diets may not have received a score, it is probable to deduce that individuals may experience side effects, especially if the calorie restriction is severe for an individual’s personalized requirements. A negative score was received if a diet’s respective resource, i.e., the book reported negative side effects such as headaches, diarrhea, nausea, lethargy, dizziness, and/or constipation. The potential maximum score was −60 points, 10 points per negative symptom and a higher negative score was therefore associated with multiple negative symptoms.

## 3. Results

The complete nutrition profile of the dietary patterns in the present study has been reported previously [3]. In summary, the healthy eating principles scored more favourably compared to popular weight loss diets. Most popular weight loss diets meal plans were deficient in multiple micronutrients. Diet quality ranged from a score of 300–460 for males and 340–530 for females (Table 1). A detailed description of the scoring categories is available in the Appendix A. A higher score indicates that the meal plan meets the Australian recommendations for food groups, macronutrient distribution, and micronutrient intake whereas a lower score is indicative of deviations from these recommendations and a less favourable nutrition profile. The AGHE (460 and 530) and Intermittent Fasting (430 and 470) had the highest scores, closely followed by the Mediterranean Diet for weight loss (MedDiet WL) (420 to 460) but no diets met the possible maximum score of 580 (Table 1). The lower diet quality scores for Keto and Paleo reflect removal of food groups, not meeting certain nutrient reference values (Appendix A), and a macronutrient distribution that differs notably from the Australian recommendations. Diets scored higher (6/7) for females, bar 8 Weeks to Wow (8WW) which scored slightly higher for males (380 versus 390) (Table 1). Furthermore, the diet with greatest variance in score between males and females was the Australian Guide to Healthy Eating for Weight Loss (AGHE WL) (460 versus 530). Average daily intake of essential vitamins and minerals can be found in Appendix A.

Behaviour change is a key component of weight loss and weight maintenance. Diets were therefore assessed for strategies to help with behaviour change. A higher score in behaviour change was a positive result indicating that the diet had strategies to encourage behaviour change and feasibility for weight maintenance. A tick denotes that the category was present in the meal plan (+5 points) and diet resource whereas the cross indicates the category was not present (−5 points). The diet 8WW had the lowest score (−5), while the AGHE WL had the highest score, meeting the maximum, 25 (Table 1). Most diets had an associated meal plan, support and resources, taught healthy food principles, and encouraged self-monitoring resulting in positive scores (Table 2). However, 8WW food restricted multiple foods and food groups, appeared to encourage short-term weight loss, and failed to meet other scoring categories, resulting in a negative score.

Similar to the behaviour change category, a higher negative scored indicated multiple negative symptoms associated with a dietary pattern. Adverse effects from popular diets are less commonly reported, especially in mainstream media where the everyday consumer may access such information and be enticed by untrustworthy health claims. A tick denotes that the negative symptom has been mentioned in the meal plan or diet resource whereas the cross indicates that the symptom was not prevalent (Table 3). The most common symptoms were associated with low carbohydrate diets including headaches, dizziness, nausea, and lethargy. Some diets were not able to be scored due to missing information, i.e., MedDiet WL and AGHE WL (Table 1). The Paleo diet met the maximum with −50, followed by the Keto diet (−40) and 8WW (−30) (Table 1). The negative symptoms attributed to Optifast were generally associated with the lower energy levels (i.e., 800 kcal–1000 kcal) due to the nature of very-low-calorie diets.

The intricacies of the cost of meal plans has previously been published [25] and discussed [18]. In general, diets that included premium, organic, or specialty products were more expensive. Meal plans cost between $93–$192 per week while the cost of a grocery shop per week ranged from $345–$624. All diets scored positively for cost with 10 points apart from Optifast (Table 3). Optifast was allocated a negative score of −10 due to exceeding the 30% threshold of income while the cost of the remaining diets were less than 30% of income [18].

## 4. Discussion

The purpose of the current study was to develop and pilot a comprehensive ranking system to assess popular diets across a range of categories including nutrition profile, cost, sustainability, and potential negative side effects. This study highlights the importance of investigating diets beyond just their nutrition profile as many popular diets can be costly, unsustainable for the long-term resulting in yoyo dieting, and leave individuals with adverse reactions such as constipation, headaches, and dizziness. No one diet has been proven to be superior despite the claims of popular diets.

### 4.1. Nutritional Adequacy

A dietary pattern that lacks variety is associated with poor nutrient intake [37,38,39,40] and is a potential risk factor for diet-related disease [41]. A key finding in this research was that multiple currently popular diets restricted dietary variety, resulting in low intakes of the five food groups which contrasts national dietary recommendations [25]. Diets which are unnecessarily restrictive and have low intakes of food groups may result in the absence of key nutrients, vitamins, minerals, and trace metals. As core food groups are excluded, these diets typically result in an increase of foods and beverages with added sugars, sodium, and unhealthy fats from discretionary sources. Diets with multiple restrictions, i.e., Paleo, Keto and 8WW, scored lower than diets that promote increased dietary variety and balanced intake from the five food groups. Previous research has also indicated that weight loss and popular diets may have unfavourable nutrition profiles with low levels of essential micronutrients integral to health [42,43,44]. Common micronutrients that may be compromised in weight loss diets are calcium, iodine, zinc, iron, and B vitamins [42,44,45]. As both the AGHE and MedDiet are evidence-based guidelines, it is unsurprising that both provided favourable scores due to their role in disease prevention and healthy ageing. For example, the MedDiet has been well researched for its role in the cognitive and cardiovascular health as well as improving mood and reducing frailty. Both the AGHE and MedDiet provide a range of antioxidants, phytochemicals, and functional foods that can modify disease risk [2].

### 4.2. Behaviour Change

The literature suggests that successful long-term weight loss and weight maintenance, individuals must make changes to lifestyle behaviour [14,46,47]. Such behaviours may include changes to diet and dietary intake, adjusting energy intake, and implementing or increasing physical activity. Without behavioural changes to diet and lifestyle choices, maintenance of weight loss and adherence is compromised [46,48]. Characteristics that threaten long-term weight loss, weight maintenance, and sustainability include unrealistic weight loss goals, restriction of food groups or drastic changes to usual dietary pattern, stress, motivation, personality, binge eating tendencies, and weight cycling [46,49,50].

Weight loss diets often do not provide the opportunity to learn and implement behaviour change and focus primarily on quick weight loss, sometimes at the expense of nutritional adequacy. Adherence is a fundamental component of weight loss success in conjunction with a caloric deficit [48]. Weight loss trials and intervention studies have been proven to be effective for the short-term; however, the likelihood of weight re-gain once individuals return to prior eating habits and dietary intake is expected if a change in behaviour does not occur [47]. Some of the researched strategies and approaches discussed in the literature regarding successful behaviour change include implementing meal plans to create structure and help to adjust to gauging energy intake based on bodily requirements, self-monitoring, and increasing knowledge and education for self-efficacy and health literacy [46,47].

Utilisation of a meal plan during weight loss can be an effective support tool [46]. In contrast, lack of meal structure and planning can lead to overeating if there is access to quantities of food that exceeds an individual’s requirements [51]. All diets in the current research had meal plans that consisted of breakfast, lunch, dinner, and snacks except for 8WW and the AGHE. The diet 8WW provided a weekly ‘food list’ that began with basic foods such as vegetables, meat and protein rich foods, dairy foods, pseudo grains (konjac rice and noodles), and cereal products. The food list provided recommendations for quantities of foods daily, i.e., 200 g/3 x a day, but no guidelines as to implementing this food list into a daily or weekly meal plan. This may result in a lack of understanding or interpretation and thus a lack of adherence or inability to follow the 8WW dietary pattern. The AGHE food pattern did not have a specific meal plan, but rather provided recipes through the Eat for Health Website.

### 4.3. Cost

Despite being a relatively wealthy country, a significant proportion of Australian’s (21%) may be food insecure, meaning that they do not have sufficient access to food [19,20]. Reasons for food insecurity relate to homelessness, competing financial priorities, low-income, household structure, and, more recently, economic shocks due to global pandemics. The current COVID-19 pandemic has resulted in twice as many Australians seeking food relief, a third of which had never experienced food insecurity prior to 2019 [19]. Expensive diets and meal plans may require a significant amount of an individual’s income, resulting in less disposable income to be spent on activities, repayments, household costs, and utilities. In the current research, Optifast and 8WW were the most expensive meal plans per week. Similarly, both meal plans required purchasing of ingredients outside of the five core food groups such as whey protein, high protein bread, and pseudo grains. Previous research has also indicated that lower carbohydrate diets like 8WW may be unaffordable due to the purchasing of uncommon ingredients including imported and organic goods [15,21]. In contrast, alternative research assessing the affordability of partial meal replacements (MR) similar to Optifast yield conflicting results, considering MR as potentially appropriate for low-income earners [52]. Healthy eating principles such as the MedDiet and the AGHE promote the inclusion of a variety of foods and food groups. These staple foods are generally non-taxable as opposed to discretionary foods such as sugar-sweetened beverages and confectionary [53,54,55]. Meal plans with an emphasis on staple non-taxable foods, coupled with purchasing home-brand items, could offer an opportunity to improve the affordability of popular diets.

### 4.4. Negative Symptoms

#### 4.4.1. Very Low-Calorie Diets (VLCD) and Meal Replacements

Optifast guidelines and research using low-calorie meal replacements reported several potential negative symptoms such as headaches, nausea, lethargy, and constipation and/or diarrhea. These, however, were related predominantly to the highest level of energy restriction (800 kcal/d) and not the level used in the present research (1200 kcal). Published literature suggests weight loss trial participants using meal replacements and VLCD may experience fatigue, dizziness, constipation, and other physiological symptoms [56] and the score was therefore applied. However, these symptoms may be more common at the highest level of energy restriction. Interestingly, there does not appear to be a correlation between high rates of non-adherence and compliance to the diets considering the symptoms [56]. Further, meal replacements and VLCD are considered safe and negative symptoms are manageable under the supervision and instruction of qualified health professionals [56,57,58]. However, for those who purchase over the counter meal replacements and do not manage their symptoms or seek advice, it may be detrimental to their health as severe symptoms have been previously reported [56,57].

#### 4.4.2. Low Carbohydrate Diets

The negative symptoms associated with Keto diets and low carbohydrate diets are commonly cited in the literature and result in low adherence [21,24,59,60,61,62,63]. Feelings of lethargy and dizziness may reflect depletion of glucose as an energy substrate and subsequent formation of ketone bodies [59]. However, symptoms and side effects of low carbohydrate diets can be more severe including elevated blood uric acid, kidney stones, dehydration, and impaired cognitive function [59,61,62,63]. Like Keto, the 8WW and Paleo resources mention headaches, lethargy, and dizziness which may relate to decreased carbohydrate intake and decreased energy intake overall. Following a gluten-free or low carbohydrate diet is detrimental to gut health for those without Coeliac disease or gluten sensitivity [21,64,65] and can result in changes to the gut microbiota as early as four weeks [66,67]. It is also possible that the lethargy and dizziness as a result of avoiding grains and cereal foods may be augmented by missing vitamins and minerals in this food group [68]. Breads, cereals, and grains are good sources of B vitamins which have a role in cognition and mood regulation (mental fatigue) [68]. Further, iron, magnesium, and other vitamins and minerals also found in some breads and cereals have a role in physical fatigue [68].

#### 4.4.3. Fasting Diets

Interestingly, the Intermittent Fasting resource did not mention potential negative side effects, explaining the missing score. Although the 2:5 protocol refers to 500–600 kcal on two fasting days and ‘normal’ eating on the remaining five days, the fasting days are still significantly energy restricted and may result in negative symptoms. Fasting and conscious calorie restriction for weight loss or religious reasons has been associated with negative symptoms and adverse effects [69,70]. VLCD and fasting can induce headaches, dehydration, fatigue, and lethargy [24,69,70]. Severe symptoms such as fever, dehydration, and hyponatremia [24] could require hospitalisation. This style of dietary pattern may be inappropriate for the elderly, young children, and those that are underweight due to the physiological risks and impairments [24].

#### 4.4.4. Healthy Eating Principles

Dissimilar to the popular diets mentioned, the MedDiet has not been reported to have negative symptoms [71,72,73]. Following a MedDiet can reduce irritable bowel syndrome symptoms, reduce pathogenic bacteria (E. coli), increase beneficial bacteria (bifidobacterial), and increase the proportions of the short chain fatty acid, acetate [71,72,73]. Similarly, negative symptoms from following a balanced diet based on the AGHE should not result in negative symptoms for most individuals. However, in a four-week dietary intervention study of 39 adult females aged 47 ± 13 years old, participants reported tiredness, low mood, hunger bloating, and headaches [21]. In this four-week intervention, a Paleo diet based on “The Paleo Diet” book was compared with an AGHE diet based on the ADG [21]. Participants in the AGHE group consumed more discretionary servings which are typically high in salt, sugar, and fat, which may have also contributed to the symptoms experienced [21]. However, for all adverse events, the Paleo comparator diet in this study had a higher proportion of participants experiencing the same negative and adverse effects [21].

## 5. Conclusions

Although innovative, the reliability and validity of the devised ranking system in the current research may require further development and adaptation to be considered a reliable tool for comparisons of dietary patterns. As noted, the negative symptoms, behaviour change, and cost categories could not be scored as accurately as diet quality due to missing information. However, this combination of data serves as a starting point to highlight the importance of a multifactorial approach which could be further developed and modified to include more categories or a range within scores to further explore the diets. Further, the discussions around low carbohydrate diets and negative symptoms may appear to be biased. However, there is proportionally more information in the literature regarding the adverse effects and symptoms experienced while following popular low carbohydrate diets in comparison to the other diets in this research, healthy eating principles, and government recommendations. Further, government recommended dietary patterns such as the AGHE and MedDiet are often disregarded by the public for weight loss despite having no dietary restrictions on foods or food groups and encouraging dietary variety. Regardless, dieting should be followed under the guidance of an appropriate health professional to minimise health deficiencies. However, given the increasing popularity of fad diets and heightened access to online nutrition information, it is important to study fad diets. Further, due to the rise of overweight and obesity within Australia, more people will be seeking weight loss diets and quality evidence is needed.

## Figures and Tables

**Table 1 nutrients-14-03414-t001:** Summary table of the ranking system results by gender and category.

	1. Diet Quality (/580)		2. Behaviour Change & Feasibility (/35)	3. Negative Symptoms (/−50)	4. Cost (/10)	Total Score	
Diet Name	Males	Females				Males	Females
Keto	300	340	5	−40	10	285	325
Paleo	290	350	5	−50	10	265	325
Intermittent fasting	430	470	10	?	10	455	495
8 Weeks to Wow	390	380	−5	−30	10	375	365
Optifast	400	420	20	−40	−10	375	395
MedDiet WL	420	460	15	?	10	455	495
AGHE WL	460	530	25	?	10	505	575

AGHE = Australian Guide to Healthy Eating; WL = Weight Loss; ? refers to missing data or a score unable to be calculated. Table 1 is the summary table with final scores from each of the four ranking system categories. Each category has been scored separately according to the methodology section. Additional information on scoring categories can be found in Appendix A.

**Table 2 nutrients-14-03414-t002:** Contribution to behaviour change category and long-term sustainability of popular weight loss diets and healthy eating principles.

Behaviour Change	Keto	Paleo	IF	8WW	Optifast	MedDiet WL	AGHE WL
Support and resources	X	X	✓	X	✓	✓	✓
Meal Plan/s	✓	✓	✓	X	✓	✓	✓
Strategies when eating out	X	X	✓	X	✓	X	✓
Encourages self-monitoring	✓	X	X	✓	✓	X	✓
Encourages nutrition knowledge	✓	✓	X	X	X	✓	✓
Promotes weight cycling ^1^	✓	X	✓	✓	X	X	X
Food restrictions ^1^	✓	✓	X	✓	X	X	X

^1^ Categories with a negative score (−5), all other categories have a positive score (+5); IF = Intermittent Fasting; 8WW = Eight Weeks to WOW; WL = Weight loss.

**Table 3 nutrients-14-03414-t003:** Negative symptoms of popular weight loss diets and healthy eating principles extracted from their respective resources.

Negative Symptoms	Keto	Paleo	IF	8WW	Optifast	MedDiet WL	AGHE WL
Headaches	✓	✓	X	✓	✓	X	X
Nausea	✓	✓	X	X	✓	X	X
Lethargy	✓	✓	X	✓	✓	X	X
Dizziness	X	✓	X	✓	X	X	X
Constipation and or diarrhea	✓	✓	X	X	✓	X	X

IF = Intermittent Fasting; 8WW = Eight Weeks to WOW; WL = Weight Loss.

## Data Availability

The datasets generated and analysed during the current study are not publicly available but are available from the corresponding author on reasonable request.

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
