# Peer review of "Developing and Piloting a Novel Ranking System to Assess Popular Dietary Patterns and Healthy Eating Principles"

_nutrients, 2022, doi:10.3390/nu14163414_

Round 1
Reviewer 1 Report
Manuscript is well written provides sufficient citation, references and contains originality.
Reviewer 2 Report
1. a novel ranking system? Please author explain “novel”. Compared with existing system, please clarify advantages and disadvantages of this novel system.
2. Table 3, Table 1? The order of Tables should be confirmed.
3. Table 1 and Table 2: please explain the reason of the data in detail.
4. The author has no relevant results or conclusions about the purpose of the paper, and it is suggested that the author further excavate the significance of the data.
5. The Results are relevant to the Discussion, or combine Results and Conclusion. Otherwise, it is difficult to understand.
Author Response
Please see attached word document.

Reviewer 3 Report
It is an interesting and meaningful work to explore potential ranking system that evaluates popular diets beyond nutritional profile. It would be better to replenish research framework and/or search process in the section of Materials and Methods. Please refer to the following literatures:
1. Li, J.M., Afsari, K., Li, N.P., et al., 2020. A review for presenting building information modeling education and research in China. Journal of Cleaner Production, 259: 120885.
2. Critselis, E., Panagiotakos, D., 2020. Adherence to the Mediterranean diet and healthy ageing: Current evidence, biological pathways, and future directions. Critical Reviews in Food Science and Nutrition, 60(13): 2148-2157.
Author Response
Please see word document attached

Round 2
Reviewer 2 Report
ok